# Towards Embedded Electrochemical Sensors for On-Site Nitrite Detection by Gold Nanoparticles Modified Screen Printed Carbon Electrodes

**DOI:** 10.3390/s23062961

**Published:** 2023-03-09

**Authors:** Anurag Adiraju, Rohan Munjal, Christian Viehweger, Ammar Al-Hamry, Amina Brahem, Jawaid Hussain, Sanhith Kommisetty, Aditya Jalasutram, Christoph Tegenkamp, Olfa Kanoun

**Affiliations:** 1Chair Measurement and Sensor Technology, Department of Electrical Engineering and Information Technology, Chemnitz University of Technology, 09107 Chemnitz, Germany; 2Analysis of Solid Surfaces, Institute for Physics, Chemnitz University of Technology, 09107 Chemnitz, Germany

**Keywords:** electrochemical sensors, electrochemical impedance spectroscopy, gold nanoparticles, electrodeposition, nitrite, potentiostat, groundwater, stability of electrochemical sensors

## Abstract

The transition of electrochemical sensors from lab-based measurements to real-time analysis requires special attention to different aspects in addition to the classical development of new sensing materials. Several critical challenges need to be addressed including a reproducible fabrication procedure, stability, lifetime, and development of cost-effective sensor electronics. In this paper, we address these aspects exemplarily for a nitrite sensor. An electrochemical sensor has been developed using one-step electrodeposited (Ed) gold nanoparticles (EdAu) for the detection of nitrite in water, which shows a low limit of detection of 0.38 µM and excellent analytical capabilities in groundwater. Experimental investigations with 10 realized sensors show a very high reproducibility enabling mass production. A comprehensive investigation of the sensor drift by calendar and cyclic aging was carried out for 160 cycles to assess the stability of the electrodes. Electrochemical impedance spectroscopy (EIS) shows significant changes with increasing aging inferring the deterioration of the electrode surface. To enable on-site measurements outside the laboratory, a compact and cost-effective wireless potentiostat combining cyclic and square wave voltammetry, and EIS capabilities has been designed and validated. The implemented methodology in this study builds a basis for the development of further on-site distributed electrochemical sensor networks.

## 1. Introduction

Chromatographic and spectroscopic methods are considered the gold standard and are well-established analytical methods for monitoring contaminants in water [1,2]. Although these methods have several advantages with respect to the analysis of real matrices, selectivity, lack of portability, complex sample preparation methods, and the need for experienced technicians [1,2] are some of the drawbacks that compel the scientific community to move toward alternative solutions. Electrochemical sensors offer a more promising solution for the detection of various contaminants. Fast response times, good sensitivity and selectivity achieved by functionalization with nanomaterials [3,4], and the possibility of miniaturization and portability for in situ monitoring without the need for trained personnel are some of the advantages of electrochemical sensors [5].

Nevertheless, studies on electrochemical sensors rarely go beyond the laboratory scale and do not reach the requirements for real field applications, such as (1) a high level of reproducibility in the fabrication of electrodes on a larger scale [6], (2) enhanced stability and lifetime of the electrode, by compensating the aging effects, e.g., due to the formation of passivation layers, and (3) the development of low-cost embedded sensor interface implementing essential measurement methods with wireless connectivity for remote monitoring. Previous literature related to the enhancement of the lifetime has focused primarily on electrode regeneration strategies [7] or using antifouling materials [8]. These methodologies involve the use of complicated setups and protocols for real-time deployment or compromise on the sensor properties and complex preparation procedures in the case of antifouling materials. Only a few studies have focused on real-time in situ monitoring, such as the work by Cuartero et al. [9] for the detection of nitrate and nitrite in seawater using submersible electrochemical sensors. Detection was achieved by adding desalination and passive acidification units for the removal of chloride and hydroxide ions before detection, which increased the complexity. Furthermore, the duration for the detection of nitrate and nitrite was approximately 1 h and 40 min, respectively, by following all the sequences before analysis [9]. In another work, algorithms such as the denoising data processing algorithm (DDPA) were implemented to correct the drift in the ion-selective sensor response towards ammonium ions. However, the developed algorithm does not consider surface fouling, which is one of the primary issues related to drift which leads to inaccurate correction capability [10].

In this paper, we focus on the electrochemical detection of nitrite by electrodeposited (gold on screen-printed carbon electrodes (EdAu/SPCE). Several aspects required for on-site monitoring such as sensing performance, stability, and lifetime are investigated. Nitrite (NO_2_^−^) is an important constituent of the nitrogen cycle and is abundantly available in food, water, and soil. It is an intermediate product formed naturally [11] during the transformation of ammonia to nitrate, or artificially by the conversion of nitrogen-converting species into nitrite [12]. It is extensively used as a food preservative, coloring agent, and chemical fertilizer [13,14], resulting in its accumulation in water and food. Although nitrite is quite useful, exceeding certain limits of exposure endangers humans and ecosystems. Excessive amounts cause algal blooms due to the eutrophication of lakes [15]. Furthermore, a high intake of nitrite can cause a reduction in hemoglobin and, consequently, the transport of oxygen in the human body [16]. This leads to the formation of nitrosamines, which are naturally carcinogenic [17]. Considering the toxicity of nitrite, several guidelines have been established regarding the permissible levels of nitrite. According to the WHO, 3 mg/L is the allowable level of nitrite in drinking water [18] and 65.2 µM for raw water. Several studies have also shown that a human intake of around 0.3–0.5 g causes poisoning and 3 g is fatal [19]. Thus, considering the health effects due to its toxicity, the development of sensors for the detection of nitrite ions is of paramount importance.

## 2. Existing Solutions in the Literature

### 2.1. Electrochemical Detection of Nitrite

Extensive research has been conducted on the electrochemical detection of nitrite by modifying the working electrode with different nanomaterials and composites owing to the high oxidation potential required for the oxidation of nitrite on bare electrodes [20]. Surfaces based on carbon nanomaterials such as carbon nanotubes [21,22], graphene [23], and graphene oxide [24,25] have been employed for the detection of nitrite. In addition, metallic and bimetallic nanoparticles, including silver [26], gold (Au) [27], and copper [28], have also been used for the detection of nitrite. Among these, Au nanoparticles are widely used because of the strong affinity of Au towards nitrite and its biocompatibility. However, Au nanoparticles are further integrated with other carbon nanomaterials [29] to enhance the surface area and sensitivity and reduce the oxidation potential of nitrite, a process that includes several complicated fabrication steps. Herein, we consider the affinity of Au nanoparticles towards nitrite [30] and demonstrate the applicability of electrodeposited Au nanoparticles for the sensitive and selective detection of nitrite. Furthermore, stability and lifetime are considered two of the major bottlenecks of electrochemical sensors for in situ monitoring. However, the aspects of stability and lifetime have not been investigated in detail previously. Herein, we take advantage of the information obtained from EIS to relate the drift in the sensor response to changes in the surface by considering different practical scenarios that have not been explored in previous studies.

### 2.2. Development of Potentiostats

With respect to the development of potentiostats, many commercial laboratory instruments are available that can implement different measurement techniques controlled over a common graphical user interface for the detection of various chemical compounds. Although these instruments offer impressive capabilities, they are limited to laboratory measurements and cannot be used for real-time in-field measurements owing to issues related to portability and their high costs.

In this regard, many researchers have developed portable potentiostat systems capable of real-time measurements of different chemical compounds. Generally, these developed systems have the capability of either voltammetry or EIS measurements, thus critically restricting the amount of information available about the surface. In addition, most potentiostats lack the capability to perform EIS measurements [31]. However, it is essential to develop a potentiostat with multi-measurement method capabilities such that the fingerprint of the electrode surface can be obtained by evaluating the information obtained from different methods. A few studies have proposed systems that can implement voltammetry and EIS methods. However, these designs use high-resolution external analog-to-digital converters (ADCs), which increases the cost and footprint of the potentiostat. For instance, in [32], a 24-bit ADC was used with an AD5933 impedance analyzer chip, thus drastically increasing the cost and footprint of the system. Although the system uses high-performance ADCs, the impedance phase measurements have high deviations compared to laboratory reference measurements. Noise issues are also a problem to overcome [29].

In [33], a 14-bit ADC was used for the measurement process, but it was observed in the cyclic voltammetry graphs that the noise level was high, and no post-processing was carried out to reduce the noise, which is crucial for accurate measurements. Moreover, EIS is implemented only on RC DUT, which does not convey the accuracy of the system when measuring chemical or biological compounds. Table 1 provides a brief comparison of previously developed potentiostats.

In this study, a simple one-step method for developing screen-printed electrodes for determining nitrite in groundwater was demonstrated. Au nanoparticles were electrodeposited on the SPCE by cyclic voltammetry, and electrochemical and physical characterizations of the EdAu/SPCE were carried out in detail. Subsequently, the developed electrode was applied to the electrochemical detection of nitrite in real samples. In addition to detection, substantial research on the stability of the electrode was carried out by considering three different approaches: (1) dry measurements, (2) immersion studies, and (3) accelerated aging for continuous monitoring. EIS was used to probe and characterize the state of the surface at every critical juncture during these studies. In addition to the development and characterization of the sensor, a potentiostat with CV, EIS, and SWV capabilities dubbed MSTStat was developed, and its efficiency was thoroughly validated by comparing it with a commercial potentiostat. The EdAu/SPCE electrode demonstrated very good analytical capability for the detection of nitrite in groundwater samples. The stability of the EdAu/SPCE electrodes was thoroughly characterized and the ability of EIS for analyzing the state of the surface at any point in time during on-site measurements was demonstrated. The work shows EdAu/SPCE for nitrite detection, with the developed potentiostat that includes EIS and SWV as a promising system for on-site monitoring of nitrite. A graphical summary of electrochemical detection of Nitrite is illustrated in Figure 1a and the components of MST Stat with its graphical user interface are shown in Figure 1b.

## 3. Materials and Methods

### 3.1. Materials

Analytical grade sodium nitrite, Au tetrachloroauric acid (99%), sodium dihydrogen phosphate monobasic (99.99%), sodium phosphate pentahydrate (99.99%), calcium sulfate (99%), and magnesium chloride (>98%) were purchased from Sigma Aldrich. Screen-printed carbon electrodes (SPCE) from ItalSens were purchased from EKT Technologies, Germany, and used without any prior treatment. The working and counter electrode material was carbon and the material for the reference electrode was silver. The pH of the solutions was adjusted with the concentration of 0.1 M hydrochloric acid and sodium hydroxide. All electrochemical measurements were performed using a PalmSens4 and the MSTStat potentiostat. The electrodes with working, counter, and reference electrodes were connected to the potentiostat through the connectors provided by the PalmSens and placed in the electrochemical cell consisting of the solutions to be analyzed. For MSTStat, the same connectors were used for connecting the electrode to the potentiostat.

### 3.2. Methods for Detection

Cyclic voltammetry (CV) was used for the electrodeposition of AuNPs. Initially, 0.01 M HAuCl_4_ was added to 10 mL of distilled water. The potential was cycled from 0 to 0.8 V at a scan rate of 0.075 V/s for five electrodeposition cycles. Before the detection, CV on EdAu/SPCE was performed to activate the gold particles and the electrodes were dried before subsequent use.

For the electrochemical characterization of the surface, 5 mM potassium hexacyanoferrate(II)/potassium hexacyanoferrate(III) (K_3_Fe(CN)_6_]^3−/4−^)was prepared in a 0.1 M KCl solution. The CV was performed from −0.4 to 0.6 V at different scan cycles and EIS spectra were obtained at 0.01 V of AC amplitude at open circuit potential and a frequency range from 0.1 to 15,000 Hz. In the case of detection, all the nitrite solutions were prepared in PBS solution with different pH for optimization and the pH with the highest response was selected for subsequent detection using SWV from 0.2 to 0.8 V at a frequency of 10 Hz. Groundwater samples were obtained from Landesamt für Umwelt, Landwirtschaft und Geologie (LfULG) Saxony, and the pH was adjusted before detection.

### 3.3. Block Diagram of MSTStat

The system design for the voltammetry and EIS measurements consists of an STM32 microcontroller with a 12-bit built-in dual DAC and 12-bit dual ADC, which is powered by a battery. The DAC generates a sine wave or square wave depending on the measurement method, including a DC offset, which is eliminated during the signal-conditioning process. The signal was then passed through the control amplifier, which excited the counter electrode of the sensor under test. The current passing through the sensor can be measured at the working electrode, which is converted to an equivalent voltage using an I-V converter. The voltage signal at the reference electrode and the output signal of the I-V converter is passed through signal conditioning amplifiers, where a DC offset voltage of 1.65 V is reintroduced. The signal is then given as an input to the two ADCs working in the simultaneous mode to avoid any phase offset during the measurement process. Digital signal processing, including a digital filter, is applied to the digitized signal, and the obtained results are then transmitted, which can be further evaluated in real-time. Figure 2 illustrates the block diagram with the components used for MSTStat.

## 4. Investigation Results

### 4.1. Electrodeposition of Gold

The electrodeposition of metallic nanoparticles is a versatile technique for obtaining electrochemical surfaces with high conductivity. Different morphologies and structures of Au nanoparticles can be obtained depending on the solvent, substrate, and electrochemical method used [42]. Herein, CV was selected because of the wealth of information obtained regarding the changes in the surface at different potentials. Furthermore, control over the electrodeposited nanoparticles can be maintained by optimizing the scan cycles and scan rates. Figure 3b shows the CV curves of the electrodeposition of AuNPs on the SPCE. The formation of reduction peaks was observed in the first scan, suggesting the reduction of Au(III) to Au(0). The reaction mechanism involved during the electrodeposition of Au is shown below [43].
(1)$$Au{Cl}_{4}^{-}+3e^{-}\to Au+4{Cl}^{-}$$

The onset of reduction in the first cycle can be observed at around 0.2 V in the first cycle and shifts to higher cathodic potentials in the subsequent scans to around 0.8 V in accordance with previous studies [44,45]. The shift in the peak to higher potentials can be attributed to the formation of a layer of AuNPs on the surface after the first cycle. This result is in accordance with thermodynamic studies that estimate the favorable growth of Au on Au as compared to graphite electrodes [46]. Furthermore, a considerable reduction in current was observed after the first scan, and thereafter, the current remained almost constant, which suggests that the active surface area of the bare electrode was covered with Au nanoparticles. This was confirmed by scanning electron microscopy images of the bare SPCE and EdAu/SPCE (Figure 3a,c). As depicted in the images, a clear difference in the surface of the bare and electrodeposited Au electrodes is observed, as a homogenous distribution of Au nanoparticles, is visible across the entire surface. The surface roughness of the developed electrodes was calculated by the equations below [47].
(2)$$Charge\left(q_{real}\right)=\frac{Area\;under\;reduction\;peak}{Scan\;rate}$$

The reduction peak in the CV with H_2_SO_4_ as the solvent was selected because the formation of the peak is related to the reduction of Au oxide, which is formed as an oxidation peak during the anodic scan [48]. The electrochemical active surface area (ECSA) was calculated from Equation (3) and subsequently the roughness factor from Equation (4).
(3)$$ECSA=\frac{Q_{real}}{Q_{theoretical}}$$
(4)$$Roughness\;factor\left(\rho \right)=\frac{ECSA}{Geometrical\;area}$$

The geometrical area of the working electrode was 0.07 cm^2^ according to the specifications, and the theoretical charge density was estimated to be 390 µC cm^−2^ for Au particles [49]. The roughness factor was calculated to be 0. 59 for the electrodeposited Au nanoparticles.

Figure 3d shows magnified SEM images of the Au nanoparticles. Variations in the size distribution and shape can be observed in the image. Non-uniformity arises due to the interaction and agglomeration of neighboring nanoparticles, leading to differences in shape and size. The histogram in Figure 3e reveals the size distribution of Au nanoparticles to be approximately 30–50 nm on the surface, and only a few extend up to 90 nm in diameter. Figure 3f shows energy-dispersive X-ray spectroscopy (EDX) measurements for elemental identification of the EdAu/SPCE surface. An overlap of the Au emission carbon peak is observed at approximately 0.2 keV and a distinctive peak is clearly visible at approximately 2.2 keV. The peaks for aluminum (Al) and sulfur (S) stem from the sample holder and the activation procedure implemented for the EdAu/SPCE by using sulfuric acid, respectively.

### 4.2. Electrochemical Characterization of SPCE and EdAu/SPCE Electrodes

The electrochemical properties and electron transfer kinetics of the SPCE and EdAu/SPCE were investigated using CV and EIS. Figure 4a,b show the CV curves of the K_3_Fe(CN)_6_]^3−/4−^redox couple on the bare SPCE and EdAu/SPCE electrodes at different scan rates, respectively. The EdAu/SPCE electrodes displayed an enhanced current compared to the bare electrode. Furthermore, the peak potential difference between the oxidation and reduction peaks was reduced considerably from 0.221 to 0.140 V for the EdAu/SPCE electrodes, suggesting the high conductivity and catalytic activity of Au nanoparticles on carbon electrodes. Figure 4c shows the linearity of the anodic peak currents versus the root of the scan rate for the modified electrodes. Linear behavior with an R^2^ value of 0.998 was observed for the EdAu/SPCE electrodes, which confirms the diffusion-controlled process involved on the surface [15].

EIS was performed to investigate the electron-transfer kinetics of the modified and unmodified electrodes. Figure 4d shows the EIS spectra of the bare and EdAu electrodes, with the inset displaying the magnified image of the spectra for the modified electrodes at high frequencies. The semicircle and linear region in the EIS spectra provide information about the charge transfer resistance and diffusion processes occurring at the interface. In this regard, as seen in Figure 4d, the semi-circle region of the modified electrodes was reduced significantly and was almost negligible compared to the bare electrodes, which implies that the deposited Au nanoparticles facilitate the charge transfer reactions occurring on the electrode [50]. A simple Randle circuit was used to fit the model to obtain semi-quantitative information about the surface. The CPE element was used instead of the double-layer capacitance to account for the surface roughness. The bare electrode typically had a high charge transfer resistance of the order of 1250 Ω, and the EdAu/SPCE electrode exhibited a lower resistance in the order of 100 Ω.

### 4.3. Benchmarking CV and EIS Methods of MSTStat

The performance of the developed MSTStat for CV and EIS was evaluated by comparing the curves obtained for the bare electrodes with those of the commercial potentiostat PalmSens. Figure 5a shows the CV curves for 5 mM of K_3_Fe(CN)_6_]^3−/4−^solution on the bare electrode by MSTStat, and Figure 5b shows a comparison of the currents obtained from both potentiostats. A qualitative analysis of the CV curves obtained with MSTStat (Figure 3a) shows similar behavior to that observed with PalmSens, with the anodic and cathodic peak current increasing with an increase in scan rate and the increase in peak separation. A more quantitative analysis can be visualized through the bar graph in Figure 5b, wherein the observed differences in the peak currents are not significant, with an error percentage of under 6%, even at high scan rates. The results demonstrate the capability of MSTStat to achieve similar performance.

Figure 5c shows a comparison of the Bode plots obtained using the potentiostats. The measurements were performed in the range of 1 Hz to 10 kHz. Deviations in phase measurements could be observed that can be attributed to the presence of parasitic capacitances on the PCB and a low signal-to-noise ratio because the applied excitation signal has an RMS magnitude of only 10 mV. The measured current in terms of voltage can be higher or lower, depending on the chemical compound and the frequency of the signal. If the current signal in terms of voltage is reduced, the measurement signal will be very close to the ADC resolution range, which leads to a decrease in the quality of the measurement and introduces deviations in the phase measurement. Due to the deviations in the phase measurements as compared to the PalmSens, the Nyquist plots for both systems were not compared in this study. However, the results show the capability of MSTStat to realize EIS measurements despite the use of low-performance components.

### 4.4. Behavior of EdAu/SPCE towards Nitrite and pH Analysis

The response of the developed EdAu/SPCE towards nitrite was investigated by CV from 0.1 to 1 V at different scan rates from 0.05 V/s to 0.150 V/s and a concentration of 500 µM. The nitrite solution was prepared in a 0.1 M phosphate buffer solution of pH 7. A clear oxidation peak is visible at around 0.6 V in Figure 6a, which increases and shifts to a higher potential with the increase in scan rate. Furthermore, the linearity in the plot of the peak current versus the logarithm of the scan rate with an R^2^ value of 0.992 indicates the dominance of diffusion-controlled processes in the oxidation of nitrite. Figure 6c shows a Tafel plot of the anodic peak potential (*E_pa_*) versus the root of the scan rate. Linear fitting of the plot leads to an R^2^ value of 0.993, and the linear regression equation is as follows:(5)$$E_{pa}=0.072\log\left(scanrate\right)+0.665$$

From Laviron’s theory, the slope obtained from peak potential versus the logarithm of the scan rate can be utilized to quantify the number of electrons transferred for the oxidation of nitrite [51].
(6)$$m=\frac{0.059}{(1-\alpha )n}$$
where *α* is the transfer coefficient and its value is 0.5, *n* is the number of electrons transferred, and m is the Tafel slope. By substituting these values, the value of *n* obtained is approximately 1.63 for the oxidation of nitrite. By considering the electron transferred as 2, the reaction mechanism involved is the electro-oxidation of nitrite to nitrate as shown in Equation (7).
(7)$$NO_{2}^{-}+H_{2}O\to NO_{3}^{-}+2H^{+}+2e^{-}$$

The number of electrons transferred was also calculated by performing bulk electrolysis and followed by coulometric analysis. A total of 0.5 mL of 250 µM nitrite solution was dropped on the screen-printed electrodes and the electrode was maintained at a constant potential of 0.5 V at which the oxidation of nitrite occurs. Figure 6d shows the charge consumed for the complete oxidation of nitrite molecules. The value of the charge consumed was quantified to be 27.8 mC. Based on Faraday’s law, represented by Equation (8), the number of electrons was determined to be 2.30.
(8)$$n=\frac{Q}{F*moles}$$

This indicates a two-electron transfer for the reaction to occur and the reaction pathway is the same as shown in Equation (7). To maximize the electrode response towards nitrite, the electrode response was evaluated at different pH values from 5 to 7. Figure 6e illustrates the increase in peak current as the pH increased to 6.5, and a drop in current was observed at pH 7. This can be attributed to the lack of sufficient protons at pH 7, which leads to a reduction in the current response of the electrode [52,53].

### 4.5. Electrochemical Detection of Nitrite by MSTStat

For electrochemical detection of nitrite, the solutions were prepared in 0.1 M PBS at pH 6.5. SWV with a voltage of 0.2 to 0.8 V and a frequency of 10 Hz was chosen for the detection of nitrite because of its ability to reduce the contribution of capacitive currents in the response. The performance of MSTStat was validated by comparing the currents obtained at all the concentrations. Figure 7a,b show the response curves for PalmSens and MSTStat, respectively, from 1 to 300 µM concentration of nitrite with their standard deviations. Figure 7a shows the increase in current with increasing concentration, which shifts to higher potentials. A qualitative comparison of the curves obtained with MSTStat revealed an almost similar response in terms of current values for the respective concentrations. However, the shift in the peak potentials for MSTStat was not monotonic and can be attributed to the interference from the parasitic capacitances (PalmSens has also parasitic capcitances, lack of shielding can cause a shift in parasitic capcitances of the developed circuit, which can be influenced due to many reasons). Nevertheless, in the context of real-time measurements, the peak current values are more significant than the peak potentials for the determination of concentrations in unknown samples. For quantitative estimation, the peak currents were evaluated using three-point measurements and similar values were observed in the calibration curves displayed in Figure 7c,d. The measurements were performed in triplicate and the standard deviations are shown in the figure. The corresponding R^2^ values that signify the accuracy of linearity were 0.996 and 0.994 for PalmSens and MSTStat, respectively. The regression equations for both calibration curves are shown in the respective plots. For statistical analysis and comparison of two linear regression lines, the F test is a versatile method for providing a statistical number for the difference between two populations [54]. In this regard, based on the standard F test at a significance level of 0.05, the value of F obtained was 2.925, and it can be inferred that the two datasets are not statistically different and the null hypothesis is valid. The limit of detection (*LOD*), based on the calibration curve obtained from PalmSens, was calculated to be 0.38 µM and 0.6 µM for MSTStat from Equation (8), where *S_b_* is the standard deviation of the blank sample and *m* is the slope of the calibration curve [55].
(9)$$LOD=\frac{3\cdot S_{b}}{m}$$

To evaluate and benchmark the performance of EdAu/SPCE towards nitrite, a comprehensive comparison of the sensor properties from previous studies is shown in the Table 2 below. The electrode developed in this study showed comparable performance to that of other electrodes previously developed in different studies. A few research studies, such as those in [40,45], developed electrodes that had linear ranges outside the permissible levels in the water. Furthermore, the electrodes were developed using complicated and multiple preparation and modification steps, such as ball milling [47], reflux heating for the preparation of zinc-based metal-organic frameworks [41], and the implementation of numerous materials for detection, such as hemoglobin, Nafion, palladium, and graphene in Ref. [40]. In this study, a simple, easy-to-fabricate, one-step strategy was implemented to develop highly sensitive electrodes for nitrite detection.

### 4.6. Selectivity Studies and Real Sample Analysis

The selectivity of electrochemical sensors is a crucial aspect to be investigated to evaluate sensor performance. In this regard, the most common interferant ions present in water, namely calcium sulfate (CaSO_4_), copper sulfate (CuSO_4_), sodium bicarbonate (NaHCO_3_), potassium nitrate (KNO_3_), and magnesium chloride (MgCl_2_) were added in 50 times excess of concentration as compared to nitrite in 0.1 M PBS solution of pH 6.5. The concentration of nitrite for the selectivity tests was 100 µM and that of the interferants was 5000 µM. The bar graph in Figure 8a shows the corresponding peak currents obtained from the oxidation of nitrite in the presence of other possible interferants with standard deviation obtained from three measurements. The plot shows no significant changes in the peak currents in the presence of high interference concentrations, with a maximum relative error of less than 12% for MgCl_2_. The results demonstrate the high selectivity achieved by the EdAu/SPCE electrodes. The reproducibility of the electrochemical sensors was quantified by fabricating 10 electrodes with the same procedure and running SVW at a nitrite concentration of 50 µM. The recorded response does not show any significant deviations across the 10 electrodes as shown from the bar graph in Figure 8b, inferring the high reproducibility of the developed electrodes.

The analytical performance of the developed electrode was addressed by evaluating its response in real samples. The response of EdAu/SPCE in groundwater samples obtained from LfULG-Saxony was characterized at different concentrations. Previous reports have demonstrated the capability of detection in real samples for one or two different concentrations. However, considering the matrix effect in real samples, the detection of a wide range of concentrations is necessary to evaluate the analytical performance of the electrode in real samples.

In this regard, the electrode response to a series of different concentrations of nitrite in groundwater was quantified. Nitrite was added to the groundwater using the standard addition method of known concentrations, and the recorded currents were compared with the currents obtained in buffer solutions. As can be observed in Figure 8c, a linear increase in the current response was observed with an increase in concentration from 50 µM to 300 µM. Table 3 shows the recovery percentages of nitrite in the groundwater samples. A very good recovery percentage, with a deviation of less than 10%, was observed for all concentrations of nitrite. Thus, it can be concluded that EdAu/SPCE has an excellent analytical capability for the detection of nitrite in complex matrices such as groundwater.

### 4.7. Stability Analysis of EdAu/SPCE

The stability of the electrode in different scenarios and environments is an important criterion for determining the applicability of electrochemical sensors for real-time detection. It is considered one of the major bottlenecks in electrochemical sensors as it restricts their deployment for on-site measurements. Existing research has focused on the development of new materials and sensors for the detection of different contaminants without a deeper understanding of their stability.

Herein, a critical and comprehensive investigation of the stability of electrochemical sensors was conducted by considering three different scenarios.

Shelf life of the sensors: In this analysis, the response of the as-fabricated electrode was recorded in triplicate at a nitrite concentration of 300 µM for 15 days. As shown in Figure 9, the changes in the peak current values did not deviate significantly over the entire 15 days, thus confirming the good shelf-life capability of the developed electrode. It should be noted that 15 days was selected for analysis, not that the electrode drifted in the response afterward, but to demonstrate that the electrode developed in this work if left in dry conditions after measurement, does not show much deviation in the response.

2.Immersion analysis in PBS: The main aim of this investigation was to force a significant drift in the response by continuous immersion of the electrode in aqueous media containing 300 µM of nitrite in PBS and characterize the state of the surface by EIS. Accordingly, SWV and EIS measurements were performed every day. EIS is selected due to its non-destructive capability in evaluating the changes on the surface. For the measurements, the electrodes were enclosed in an airtight bottle containing a 300 µM concentration of nitrite and the responses were recorded in the same solution. A significant reduction in the peak current and a shift in the curves to higher potentials was observed as seen in Figure 10b. For EIS, the measurement was performed at a DC potential of 0.5 V, with an AC amplitude of 0.01 V and a frequency range from 0.1 to 15,000 Hz.

Figure 10a shows the EIS spectra obtained after the SWV measurements were taken every day. Qualitative inspection of the spectra revealed significant changes as the number of days increased. The inset of Figure 10a shows the equivalent circuit used to fit the spectra. In this regard, from days 1 to 5, the equivalent circuit elements include a solution resistance (R_sol_) in series with an RC element and CPE element to model diffusion. The CPE was considered instead of the capacitor, owing to the imperfections associated with the double layer. Furthermore, CPE was also selected to model the diffusion, due to the surface roughness and inhomogeneities. It can be observed that only one RC element was sufficient to fit the equivalent circuit for the spectra obtained until the 5th day. However, as shown, an additional RC element was required to fit the spectra for days 7 and 9. The physical explanation for the 2nd RC element can be related to the formation of a thin passivating layer on the surface of the electrode owing to fouling, which creates an additional barrier for ions to diffuse to the surface of the electrode [66].

Nevertheless, the quantitative values of the fitted parameters provided valuable information about the state of the surface. The charge transfer resistance (R_1_) increased from 2038 Ω on day 1 to 3052 and 3979 Ω on days 3 and 5, respectively. Through SWV measurements, a substantial reduction in peak current and shift in peak potentials to higher values was observed, as shown in the bar graph in Figure 10c. In addition, the exponent value of CPE_2_ increased from 0.627 to 0.730, suggesting an increase in the homogeneity of the underlying surface. This could be related to the growth of the fouling layer from small and discontinuous particles to a semicontinuous layer on the surface on day 5. However, from day 7, the fitted equivalent circuit consisted of two RC elements, wherein the RC element at higher frequencies arose owing to the continuous fouling layer formed on the surface, and the other RC element at low frequencies was due to the charge transfer resistance. The value of R_1_, or the resistance of the fouling layer, increased from 1918 to 2066 Ω from days 7 to 9. Similarly, an increase in the exponent value of CPE_1_ from 0.966 to 0.990 was observed from days 7 to 9, suggesting a highly uniform and homogeneous fouling layer formed on the surface. One of the points worth noting is that the increase in the resistance value for day 9 is quite low and can be complemented very well by a very small reduction in the peak current from day 7 to 9, as observed in Figure 10c.

3.Accelerated tests for continuous monitoring: The main aim of these experiments was to test the stability of EdAu/SPCE by pushing it to its limits. In this regard, CV is considered a destructive electrochemical method because the electrode is subjected to potential changes within a short time, and thus, has the highest probability of deteriorating the sensor surface. Herein, to manifest excessive stress on the electrode surface, after the initial 25 SWV scans in presence of nitrite (300 µM), the electrode was subjected to 20 cycles of CV from 0 to 1 V at a very high scan rate of 0.25 V/s and very high concentration of nitrite (2000 µM) before the next SWV measurement.

Figure 11 shows the EIS and SWV measurements obtained after the initial 25 SWV scans in the presence of the same concentration of nitrite. As depicted in Figure 11a, the current decreased substantially after the first and second cycling by CV at high scan rates and concentrations from 111 to 87 and 71 µA, respectively. However, after the third CV cycling, the reduction in current was lower (66 µA). To further elevate the deterioration of the electrode, 100 CV cycles with the same parameters were performed on the same electrode. Nevertheless, the electrode was able to sustain the high stress by showing only a reduction of approximately 3 µA in the current (63 µA). The results demonstrated that after 160 cycles of high stress on the electrode, an approximately 50% reduction in the current was observed. Figure 11b,c show Nyquist and Bode plots recorded after each cycling phase on the electrode. The parameters selected for EIS were similar to those of the immersion tests and were measured in the presence of nitrite in the PBS solution. A considerable change in the spectra was observed from the Nyquist plot, with the charge transfer resistance changing from 5237 Ω without any cycling to 7021 Ω after 100 cycles of scanning.

Apart from the Nyquist plot, a consistent increase in the phase angle maximum was observed after every cycling phase, which suggests an increasing tendency in the formation of either a layer or deterioration of the surface [67].

The accelerated tests demonstrate the high stability of the electrodes under high external stress, such as CV, at high scan rates and concentrations. However, the electrode showed a drastic reduction of current in the immersion tests due to the formation of a passivation layer on the surface. However, the shelf life of the developed electrode was very good, with minimal deviations in the response across 15 days.

## 5. Conclusions

The realization of electrochemical sensors for field application is associated with several challenges. In addition to achieving the desired sensitivity and selectivity, the fabrication of the sensors should be uncomplicated and achieve high reproducibility to enable mass production. The stability of the sensors is one of the crucial aspects which must be investigated.

In this paper, we propose a one-step modification of screen-printed electrodes by electrodeposited Au nanoparticles with an average particle size of 30–50 nm. The developed sensors were capable to achieve highly sensitive and selective detection of nitrite with a very low limit of detection of 0.38 µM in a linear range of 1–300 µM. The electrode showed excellent analytical capability for the detection of nitrite in groundwater with a maximum error of around 10% for all concentrations. Considering the real-time applicability of electrodes, stability analysis is one of the issues seldom investigated. An extensive and in-depth analysis of the stability of the electrode was carried out by evaluating the response of the electrode in different relevant scenarios. The extent and effect of fouling on the electrode response were thoroughly characterized and quantified by fitting the EIS spectra obtained in parallel to SVW measurements from day 1 to 9. Indirect information about the growth of a fouling layer during the immersion tests was obtained by evaluating the EIS spectra. An additional RC element was required to fit the spectra from day 7 onwards owing to the onset of the formation of a fouling layer. In accelerated aging tests, the electrodes showed excellent stability even under extreme stress cycling (160 cycles) by CV at a high scan rate and high concentration of nitrite. By considering the number of cycles (160) for accelerated aging and the reduction in current (50%), it can be concluded that EdAu/SPCE has high stability even under extreme stress. A potentiostat dubbed MSTStat was developed with the capability of CV, EIS, and SWV measurements and validated. A cost-effective potentiostat was able to show comparable performance as that of PalmSens for all the measurement methods.

The research carried out within this paper provides a new direction to go beyond the proof of concept of electrochemical sensors in laboratories to real-time on-site applications. The importance of quantifying the drift in sensor response was highlighted and future research must emphasize more on compensating for the drift or developing materials that display high stability. In addition, for on-site monitoring, the focus must also be on the development of cost-effective embedded solutions with the capabilities of performing multiple measurement methods and integrated with the wireless transmission of data.

## Figures and Tables

**Figure 1 sensors-23-02961-f001:**
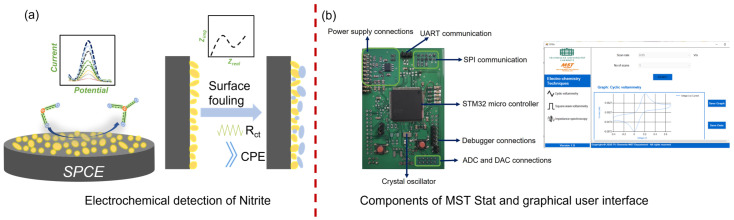
Graphical abstract of (**a**) Developed electrode for determination of nitrite and its surface characterization, and (**b**) The components of MSTStat and images of the graphical user interface.

**Figure 2 sensors-23-02961-f002:**
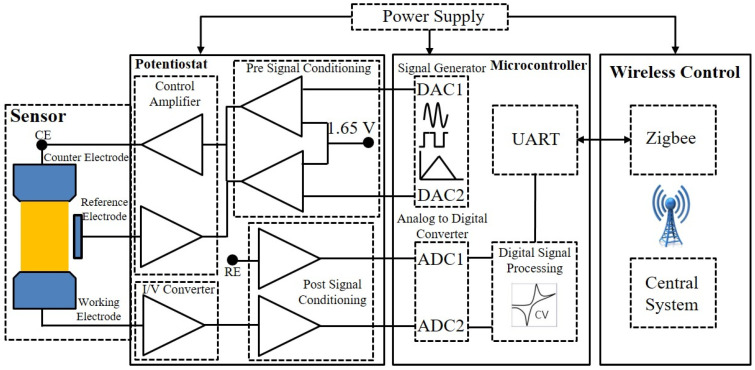
Block diagram with the components for MSTStat.

**Figure 3 sensors-23-02961-f003:**
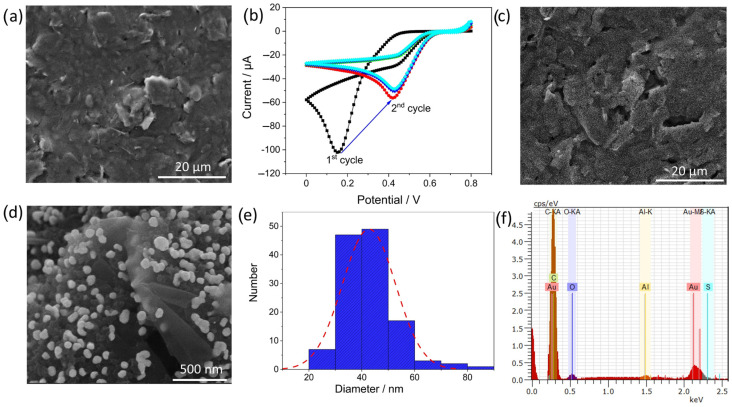
SEM images of (**a**,**c**) bare SPCE and SPCE after electrodeposition, (**b**) shows the electrochemical deposition of Au nanoparticles by CV in 0.01 M HAuCl_4_ with distilled water as supporting electrolyte and scan rate of 0.075 V/s for five cycles, (**d**) magnified SEM image showing the distribution of Au nanoparticles on the surface, (**e**) Histogram for the size distribution of Au on the surface evaluated from ImageJ and (**f**) shows the corresponding EDX spectra of EdAu/SPCE.

**Figure 4 sensors-23-02961-f004:**
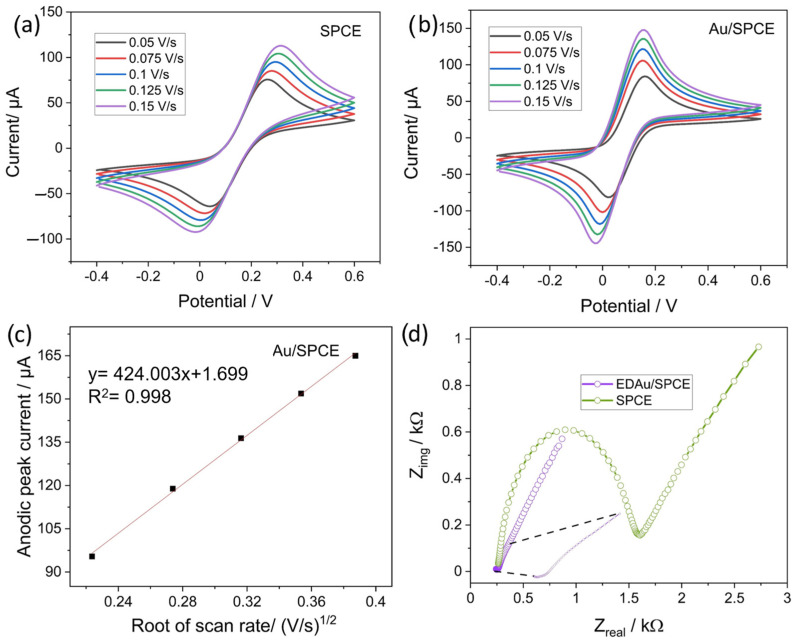
(**a**,**b**) show the CV curves at different scan rates in 5 mM K_3_Fe(CN)_6_]^3−/4−^solution for the SPCE and EdAu/SPCE electrodes, (**c**) depicts the linearity between the anodic peak current and the root of the scan rate for the EdAu/SPCE electrode, and (**d**) is the Nyquist plot of the SPCE and EdAu/SPCE electrodes.

**Figure 5 sensors-23-02961-f005:**
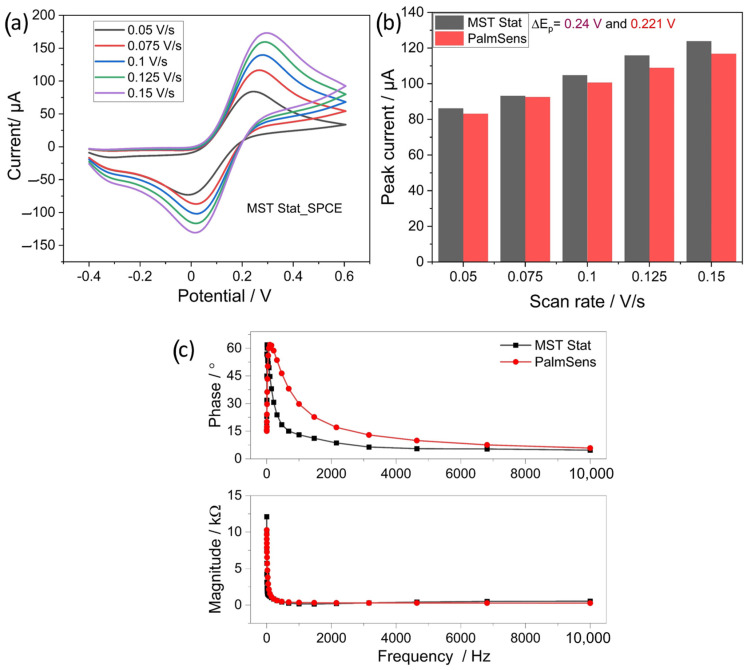
Validation of the developed MSTStat with PalmSens by (**a**) measurement of the CV of 5 mM K_3_Fe(CN)_6_]^3−/4−^ solution on bare electrodes at different scan rates, (**b**) bar graph depicting the measured currents from both the potentiostat and (**c**) depicts the phase and magnitude plots of the bare electrode by PalmSens and MSTStat.

**Figure 6 sensors-23-02961-f006:**
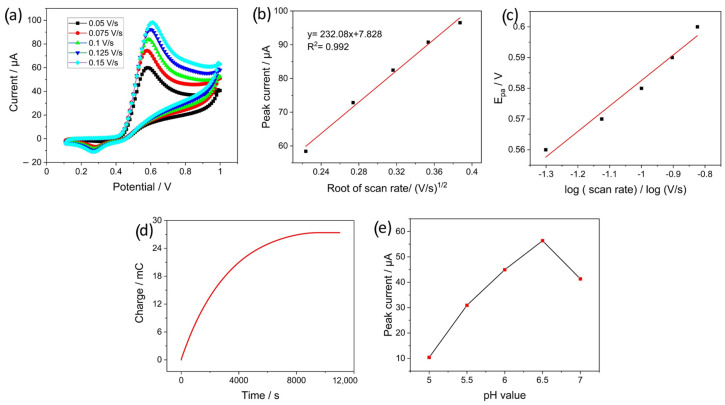
CV curves for (**a**) investigation of the oxidation mechanism of nitrite on EdAu electrode, (**b**) plot of peak current from CV versus root of scan rate; (**c**) Tafel plot obtained from the peak potential from CV at different scan rates, (**d**) the charge consumed during bulk electrolysis for 250 µM of nitrite, and (**e**) the role of pH in the electrocatalytic oxidation of nitrite.

**Figure 7 sensors-23-02961-f007:**
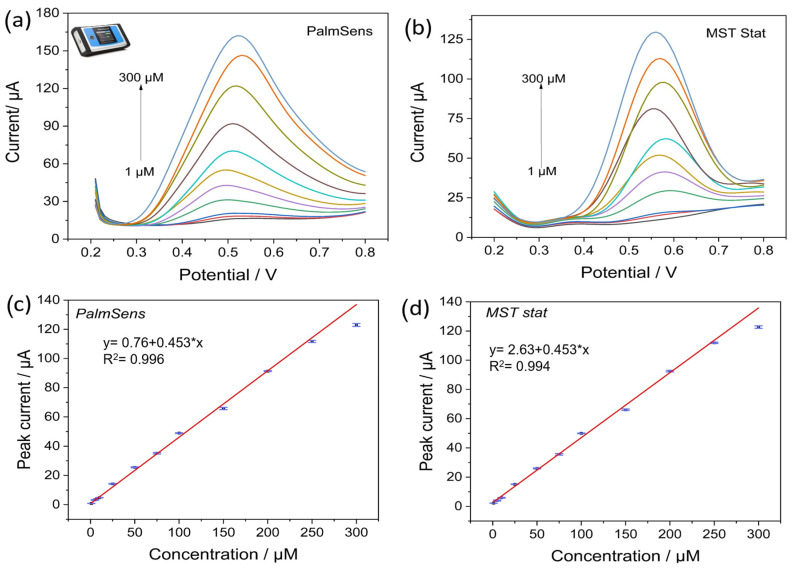
Square wave voltammetric curves at different concentrations for (**a**) PalmSens, (**b**) MSTStat and (**c**,**d**) the calibration curves with the linear regression equation for PalmSens and MSTStat respectively.

**Figure 8 sensors-23-02961-f008:**
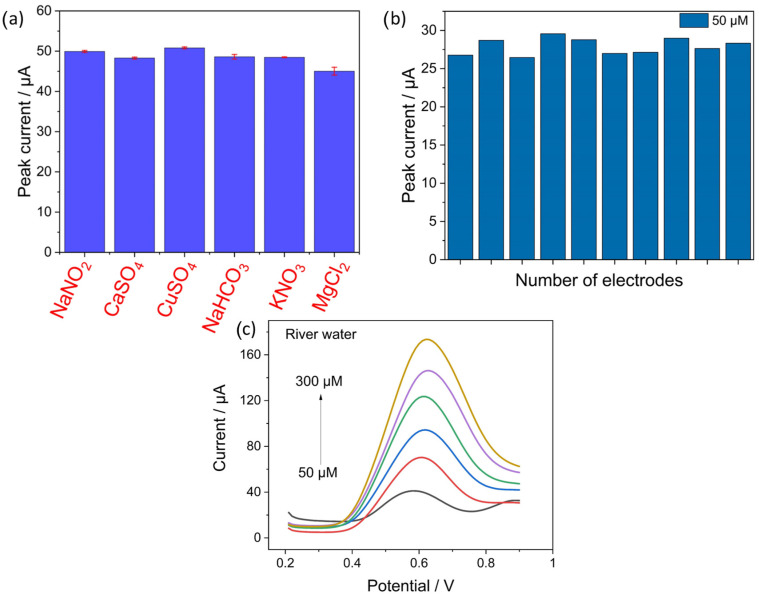
(**a**) Selectivity studies of EdAu/SPCE electrode with possible interferences of concentrations 50 times higher than nitrite, (**b**) reproducibility of ten electrodes towards nitrite of 50 µM concentration and (**c**) shows the response of the developed electrode in groundwater.

**Figure 9 sensors-23-02961-f009:**
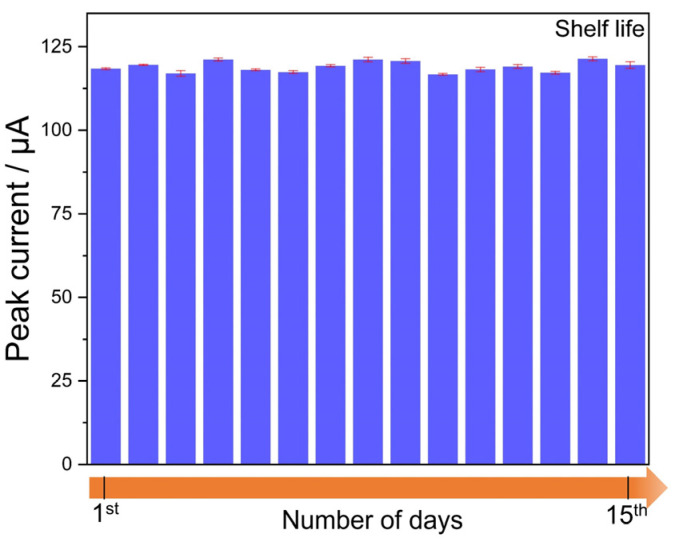
Bar visualization of minimal deviation in EdAu/SPCE towards nitrite over 15 days.

**Figure 10 sensors-23-02961-f010:**
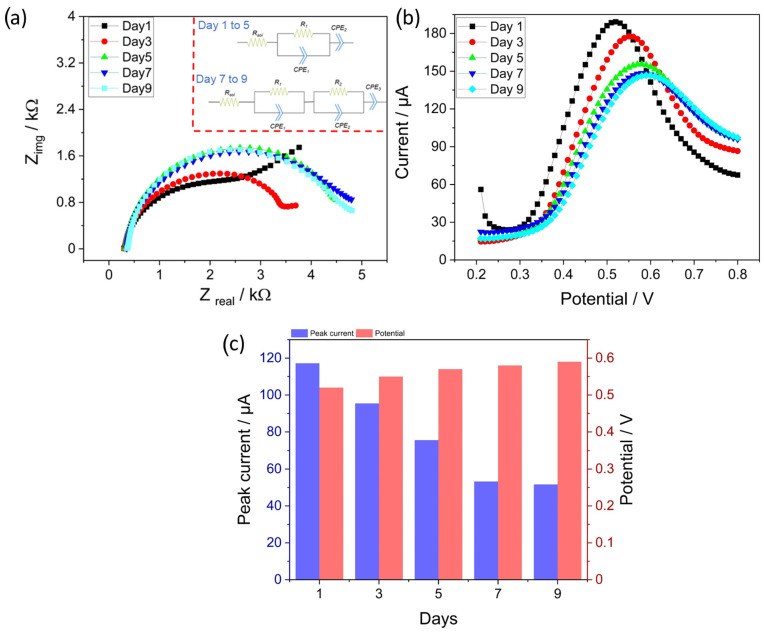
(**a**) EIS spectra of EdAu/SPCE electrode with respect to the days and the inset shows the equivalent circuit, (**b**) shows the corresponding SWV measurements, and 10 (**c**) Bar graph showing the variation in peak potential and peak currents obtained from SWV.

**Figure 11 sensors-23-02961-f011:**
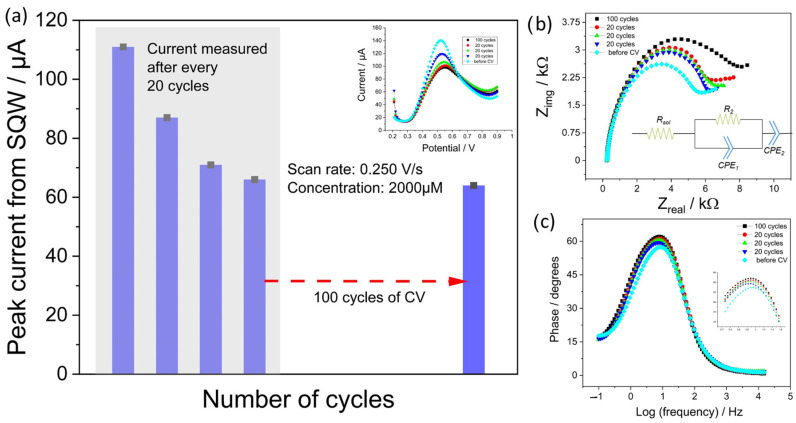
(**a**) bar graph depicting the decrease in peak current after every cycling phase and the inset shows the SWV curves, (**b**,**c**) shows the Nyquist and phase plots of the electrode respectively and the inset in Figure 11b shows the equivalent circuit model implemented for fitting the spectra.

**Table 1 sensors-23-02961-t001:** Comparison of the specifications of different potentiostats from the literature.

Ref.	Methods Implemented	Interface	Potential Range in V	Resolution in Bit
[32]	CV, DPV, EIS	Bluetooth	−1.65 to 1.65	24
[33]	CV, CA, EIS, SWV	-	−12 to 12	14
[34]	CV, LSC, CA	USB	−2.5 to 2.5	-
[35]	CV	-	-	16
[36]	CV, LSV, CA, CC	USB	−1.5 to 1.5	16
[37]	CV, LSV, CA	Bluetooth, WiFi	−1.5 to 1.5	12
[38]	CV, LSV, CA	Bluetooth	−1.5 to 1.5	-
[39]	CV, CA	NFC	−0.8 to 0.8	-
[40]	LSV	USB	-	24
[41]	DPV	WiFi	−0.6 to 0.6	12
**This work**	**CV, SWV, EIS**	**Bluetooth, Zigbee, USB**	**−1.65 to 1.65**	**12**

CV: Cyclic voltammetry, LSV: Linear sweep voltammetry, CA: Chronoamperometry, DPV: Differential pulse voltammetry, CC: Chronocoulometry, USB: Universal serial bus, NFC: Near field communication.

**Table 2 sensors-23-02961-t002:** Performance comparison of the electrodes developed in this study with other electrodes from the literature.

Electrode	Technique	*LOD* (µM)	Linear Range (µM)	Ref
AgMC-PAA/PVA/SPCE	Amperometry	4.5	2–800	[56]
CO_3_O_4_/RGO	Amperometry	0.14	1–380	[57]
Hb-Nafion/Pd-Gr/CILE	Voltammetry	30	600–61,000	[58]
Au/Zn-MOF/GCE	CV	1	5–65,000	[59]
Nano-Au/P3MT/GCE	Amperometry	2.3	10–1000	[52]
PEDOT-AuNps/GCE	Amperometry	0.1	3–300	[60]
Co nanoflowers/CPE	Amperometry	1.19	100–2150	[61]
LIG/F-MWCNT-AuNps	Voltammetry	0.9	10–140	[62]
CSPE/AuNPs-PEI	Voltammetry	0.0025	0.01–4	[63]
AgNP/GCE	Amperometry	1.20	10–1000	[64]
Ni/MOS_2_/GCE	Voltammetry	2.48	5–800	[65]
**EdAu/SPCE**	**Voltammetry**	**0.38**	**1–300**	**This work**

**Table 3 sensors-23-02961-t003:** Recovery and analysis of nitrite in groundwater by EdAu/SPCE.

Added Concentration (µM)	Current Recorded (µA)	Recovery (%)	RSD (%) (*n* = 3)
50	23.54	92.13	2.19
100	51.406	105.08	0.91
150	71.76	108.95	3.58
200	86.52	94.72	4.71
250	101.2	90.61	3.26
300	117.4	95.45	5.78

RSD: Relative standard deviation.

## Data Availability

Data is not made available.

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
