# Peer review of "Towards Embedded Electrochemical Sensors for On-Site Nitrite Detection by Gold Nanoparticles Modified Screen Printed Carbon Electrodes"

_sensors, 2023, doi:10.3390/s23062961_

Round 1

Reviewer 1 Report

The manuscript is a research article entitled “Towards embedded electrochemical sensors for on-site nitrite detection by gold nanoparticles modified carbon electrodes”  Electrochemical sensors have attracted a great deal of attention for their applications. Before accepting it for publication, the following points need to be addressed:

(1) “Screen-printed” should be included in the title of the manuscript.

(2) It is essential to mention the purity of the chemicals in the materials section.

(3) Ed must be identified in the abbreviation (EdAu/SPCE) to facilitate reader understanding.

(4) Lines 163-165: More details about screen-printed carbon electrodes (SPCE) from ItalSens should be included, such as the product number and the type of reference and counter electrodes. Also, describe the electrochemical cell used for measurements.

(5) Line 174: ferri-ferro cyanide should be replaced by the corrected name and structure a like mixture of potassium hexacyanoferrate(II)/ potassium hexacyanoferrate(III), K3Fe(CN)6]3−/4−.

(6) Lines 170-171: “The potential was cycled from 0 to 0.8 V at a scan rate of 0.075 V/s for five electrodeposition cycles.” Is the potential set from 0 to 0.8V or from 0.8 to 0V?

(7) Line 228: It is necessary to correct the chemical formula of H2SO4.

(8) In the first mention, ECSA must be identified.

(9) In the caption of Fig. 3B, you must mention the experiment parameters, such as the concentration of the HAuCl4, supporting electrolyte,  scan rate, and the number of cycles.

(10) It is recommended to perform bulk electrolysis with coulometric analysis to determine the number of electrons transferred during nitrite oxidation. After calculating the net charge consumed, Faraday's law is applied.

(11) Moderate editing of the English language and style is required for the whole manuscript.

As of now, I need to receive a response from the authors before making a decision.

Reviewer 2 Report

1. Check this manuscript to avoid English language errors carefully, especially grammar and spelling errors, superscripts and subscripts, etc.

2. There are some formatting issues, e.g., equations are not well-aligned.

3. Please cite reference papers for “Linear behavior with 264 an R2 value of 0.998 was observed for the EdAu/SPCE electrodes, which confirms the dif-265 fusion-controlled process involved on the surface.”

4. In a few figures, if your x-axis is “root of scan rate”, then your unit is wrong in the x-axis label.

5. Why PH test stopped at PH7? What about in alkaline solution?

6. Why nitrite has different oxidation potential for PalmSens and MTS Stat since the same sensing electrode were used?

Round 2

Reviewer 1 Report

All of my questions and comments were answered by the authors. The manuscript has been revised to incorporate all suggestions. The manuscript is suitable for publication in Sensors journal as it is consistent with its content.

Reviewer 2 Report

1. Typos are still remaining in the manuscript. e.g., "THe" at line 348, "ml" at line 178, "HAuCl4" at line 260, line 279, line 229, and many more.

2. Captions are not in the same formatting, please fix it.
